# Validation of RNA Extraction Methods and Suitable Reference Genes for Gene Expression Studies in Developing Fetal Human Inner Ear Tissue

**DOI:** 10.3390/ijms25052907

**Published:** 2024-03-02

**Authors:** Claudia Steinacher, Dietmar Rieder, Jasmin E. Turner, Nita Solanky, Shin-ya Nishio, Shin-ichi Usami, Barbara Hausott, Anneliese Schrott-Fischer, Jozsef Dudas

**Affiliations:** 1Department of Otorhinolaryngology, Medical University Innsbruck, 6020 Innsbruck, Austria; claudia.steinacher@i-med.ac.at (C.S.); annelies.schrott@i-med.ac.at (A.S.-F.); 2Institute of Bioinformatics, Medical University Innsbruck, 6020 Innsbruck, Austria; dietmar.rieder@i-med.ac.at; 3Biosciences Institute, Newcastle University, Newcastle upon Tyne NE1 4EP, UK; jasmin.turner@newcastle.ac.uk; 4UCL Great Ormond Street Institute of Child Health, University College London, London WC1N 1EH, UK; nita.solanky@ucl.ac.uk; 5Department of Hearing Implant Sciences, Shinshu University School of Medicine, Matsumoto 3-1-1 Asahi, Nagano 390-8621, Japan; nishio@shinshu-u.ac.jp (S.-y.N.); usami@shinshu-u.ac.jp (S.-i.U.); 6Institute of Neuroanatomy, Medical University Innsbruck, 6020 Innsbruck, Austria; barbara.hausott@i-med.ac.at

**Keywords:** human fetal inner ear, reference gene, RNA extraction, RNA expression, *TECTA*, *OTOF*

## Abstract

A comprehensive gene expression investigation requires high-quality RNA extraction, in sufficient amounts for real-time quantitative polymerase chain reaction and next-generation sequencing. In this work, we compared different RNA extraction methods and evaluated different reference genes for gene expression studies in the fetal human inner ear. We compared the RNA extracted from formalin-fixed paraffin-embedded tissue with fresh tissue stored at −80 °C in RNAlater solution and validated the expression stability of 12 reference genes (from gestational week 11 to 19). The RNA from fresh tissue in RNAlater resulted in higher amounts and a better quality of RNA than that from the paraffin-embedded tissue. The reference gene evaluation exhibited four stably expressed reference genes (*B2M*, *HPRT1*, *GAPDH* and *GUSB*). The selected reference genes were then used to examine the effect on the expression outcome of target genes (*OTOF* and *TECTA*), which are known to be regulated during inner ear development. The selected reference genes displayed no differences in the expression profile of *OTOF* and *TECTA*, which was confirmed by immunostaining. The results underline the importance of the choice of the RNA extraction method and reference genes used in gene expression studies.

## 1. Introduction

Today, there is a high interest in understanding the RNA-based gene expressions that reveal the activation or inhibition of genes together with their corresponding up- or down-regulated pathways. Comprehensive gene expression studies depend on high-quality RNA, which can be used for real-time quantitative polymerase chain reaction (RT-qPCR) and next-generating sequencing (NGS) [1,2,3]. It is of major interest to discover the genes that are regulated during inner ear development in humans [1,4,5]. A concise gene expression analysis in the human developing inner ear is required to better understand the function of deafness-related genes, as well as for the development of future efficient therapies in addition to cochlea implants [6].

In the year 1988, Rupp and Locker isolated RNA from formalin-fixed, paraffin-embedded (FFPE) tissue for the first time [7]. In 1997, Lee et al. [8] extracted RNA from FFPE stored samples of adult human temporal bones for the first time. Following these prominent and early studies, several techniques for extraction have been developed to improve the quality and amount of RNA generated from FFPE tissue. The issues associated with RNA extraction from FFPE tissue are RNA degradation and chemical modification during the fixation steps that lead to molecular changes and limitations in the availability of RNA [3,9,10,11,12,13]. RNA can also be extracted from non-fixed, fresh frozen tissue stored in RNAlater solution, which can improve the RNA in terms of its quality and amount [3,10,11,12,14].

A commonly used and sensitive method used to quantify the level of mRNA expression in specific target genes is RT-qPCR [15,16]. However, the initial sample quantity and quality, as well as the recovery and integrity of RNA, influence the results of gene expression analysis [17,18]. Therefore, it is important to apply an efficient RNA extraction method and to select the most reliable reference genes for data normalization [19]. Glyceraldehyde-3-phosphate dehydrogenase (*GAPDH*) or β-actin (*ACTB*) are routinely used as reference genes. These reference genes are frequently adopted from the literature without validating their expression stability under specific experimental conditions. Suitable reference genes require a constitutive, non-regulated, stable expression to be used in gene expression studies [15,16,18,19].

To examine the influence of different reference genes on the expression level of target genes, we chose two genes that play an important role in inner ear tissue. One is otoferlin, also known as *OTOF*. This gene is an important factor in the calcium-dependent fusion of synaptic vesicles in hair cells [20,21,22]. *OTOF* knockout mice reveal reduced calcium-dependent exocytosis in the inner hair cells. A complete disruption of the *OTOF* gene leads to the hereditary hearing loss DFNB89, while mutations can lead to profound hearing impairment [23,24,25,26]. The second target gene is alpha-tectorin, known as *TECTA*. *TECTA* is a large, flexible, non-collagenous glycoprotein. The protein is a component of the tectorial membrane, which is an extracellular matrix that covers the organ of Corti [27,28,29]. The tectorial membrane comes in contact with the stereocilia bundles of the sensory hair cells and plays a role in the induction of stereocilia movement. The movement of the stereocilia triggers hair cell depolarization and transforms the changes in the acoustic pressure into an electrical signal. Defects in the *TECTA* gene cause hearing impairment and prelingual deafness [28,29,30,31].

Gene therapy offers new treatment options for hearing loss due to the confined space and the surgical accessibility of the inner ear. Molecular tools like gene replacement, antisense-oligonucleotides, RNA interference and gene editing via the CRISPR/Cas-system allow the correction of genetic hearing impairment. Gene replacement therapy using the *OTOF* coding sequence delivered by adeno-associated virus (AAV) vectors was successfully applied in *OTOF*-deficient mice and restored hearing postnatally and at adult stages [32,33,34]. Recent data provide evidence for the clinical safety and efficiency of AAV-mediated *OTOF* gene therapy in humans [35]. 

In this study, we evaluated different RNA extraction methods for human fetal inner ear tissue, due to its importance in developmental and regenerative studies. Limitations associated with its postmortem changes and poor accessibility due to its location inside the temporal bone differ from animal experiments, which have controlled inner ear extraction that is not hidden inside a dense bone and can immediately be processed after sacrifice. Furthermore, we analyzed the most stable reference genes for gene expression studies, which can be used for future studies of inner ear gene expression.

## 2. Results

### 2.1. Comparison of Different RNA Extraction Methods for Human Foetal Inner Ear Tissue

The analysis of human fetal inner ear specimens requires a suitable extraction method that provides a sufficient amount (~500 ng input material) of RNA with good quality (RIN > 7) to generate RNASeq libraries and to run RT-qPCR experiments. Protocols for RNA extraction from fetal inner ear tissue are lacking and RNA isolation from temporal bones is challenging. Therefore, we tested two different methods for paraffin-embedded samples: the High Pure FFPET RNA Isolation Kit and RecoverAll^TM^ total Nucleic Acid Isolation Kit for FFPE. For fresh frozen tissue stored in RNAlater solution, we utilized RNA isolation with Ambion Trizol and a combination method involving RNA isolation with Ambion Trizol and the RNeasy Micro Kit. 

The four RNA extraction methods resulted in extracted RNA of different quantities and qualities. The Trizol method produced the highest RNA amount of 1668 ng ± 135, followed by Trizol/RNeasy with 1424 ng ± 120 and FFPE RecoverALL with 3.7 ng ± 1.0. By contrast, no RNA was obtained with FFPE High Pure (Figure 1A). Since it is very important for any further analysis that the RNA is not degraded, we performed RNA integrity number (RIN) analyses using the Agilent Bioanalyzer, which produces an electropherogram profile of capillary gel electrophoresis for each sample [36]. The results showed that Trizol/RNeasy retrieved most intact RNA, with a RIN around seven to nine (Figure 1B) and clearly visible peaks at 18S rRNA and 28S rRNA (Figure 1C). Trizol showed a more widely distributed RIN range between two to nine, and FFPE RecoverALL resulted in poor RIN values of around two. For further RT-qPCR and NGS analyses, we combined the Trizol and RNeasy methods.

### 2.2. Reference Gene Validation for Human Foetal Inner Ear Gene Expression

A gene expression analysis of human inner ear tissue requires appropriate reference genes. So far, suitable reference genes for the developing human inner ear are lacking. Therefore, we validated twelve different reference genes for their suitability in gene expression studies of the human fetal inner ear.

The statistical assessment of the expression stability of reference genes was performed with the Brown-Forsythe one-way ANOVA for RT-qPCR and with DESeq2 for NGS data. We calculated the normalized fold change compared to gestational week (GW) 11, which was used as the experimental calibrator (Figure 2). Two reference genes (*B2M*, *GUSB*) showed a highly significant variation (Table 1) in the RT-qPCR during the investigated gestational weeks. More moderate variations were observed in seven out of the twelve tested reference genes (*ACTB*, *GAPDH*, *HPRT1*, *TBP*, *TUBB*, *UBC*, *YWHAZ*), and three genes displayed no significant variation (*PPIA*, *RPLP*, *RRN18S*) during the GWs. The RT-qPCR and RNAseq results are plotted for graphical visualization together in Figure 2. Some references genes showed almost the same expression profile with both methods, while others displayed a different expression. A similar expression profile over the time with RNAseq and RT-qPCR was observed for five reference genes (*ACTB*, *HPRT1*, *PPIA*, *RPLP* and *TUBB*). *B2M*, *GAPDH*, *GUSB*, *TBP*, *UBC* and *YWAHZ* displayed a relatively stable expression in RNAseq, but with RT-qPCR, some peaks were visible in the expression profile. A different expression profile for RT-qPCR and RNAseq was observed for RRN18S, which showed an increase in the RT-qPCR and a decrease in the RNAseq expression data. Figure 3 shows that most of the average mean threshold cycles (Cts) for the reference genes are between 20 and 30 cycles, though *RRN18S* has a higher mean expression at about cycle 10 and *ACTB* shows a higher variability in the standard deviation (SD). 

To evaluate the expression stability of the reference genes, we used four common statistical algorithms for RT-qPCR analyses: NormFinder version 0.953 [17], mean ± standard deviation (SD), coefficient of variation (CV) analysis [16], and pairwise ΔCt [18] (Table 2). The NormFinder algorithm calculates the expression stability using the variation and intragroup variation parameters. The stability score, which represents the systematic error of practical measurement (like as standard error), is denoted by the S value [17,19]. NormFinder revealed that *HPRT1* is the most stable gene, with minimal variations across the developmental time points. The mean ± SD identified that *B2M* was the most stable gene at all investigated stages. In the CV analysis, *HPRT1* was the most stable reference gene. The CV is calculated as the ratio of the SD to the mean from the linearized Ct values across the samples and is expressed as a percentage. A lower CV value indicates a higher stability [16]. The pairwise ΔCt identified that *GAPDH* was the most stable gene. This algorithm calculates the stability as the average SD of the Ct value differences that the gene shows compared to other genes [18]. 

Taken together, the four genes *B2M, GAPDH, GUSB* and *HPRT1* were found to be the most stable and useful reference genes for expression analysis in human fetal inner ear tissue.

### 2.3. Influence of Selected Reference Genes on Expression Profile of Target Genes

As a final step, we investigated the effects of different reference genes on the expression of the target gene profile. Therefore, we chose the most stable reference genes, namely *B2M*, *GAPDH*, *GUSB* and *HPRT1*, for further analysis. As an example of target genes, we selected the two well-known deafness genes *OTOF* and *TECTA*. 

The four reference genes *B2M*, *GAPDH*, *GUSB* and *HPRT1* had no influence on the expression profile of *OTOF* and *TECTA* (Figure 4). Some differences were identified in the level of expression. *OTOF* normalized to *B2M* and *HPRT1* showed no significant changes between GW11 and GW19, whereas the genes *GAPDH* and *GUSB* exhibited statistical differences. In comparison, *TECTA* showed significant changes between GW11 and GW19 with all four reference genes. To confirm the expression profiles obtained by RT-qPCR, we performed immunohistochemical staining. The immunostaining of *OTOF* and *TECTA* showed a visible increase in the intensity and pattern during development. Figure 5A,B show the lack of or low immunostaining of *OTOF* and *TECTA* at GW12, which correlates with the RNA expression. The immunoreactivity of *OTOF* and *TECTA* increased until GW19, similar to the RNA levels (Figure 5). Thus, the immunostaining for *OTOF* and *TECTA* confirms the results from the RT-qPCR at the protein level. 

## 3. Discussion

During human development, the inner ear undergoes dramatic changes in morphology that in turn lead to a well-functioning inner ear [37,38,39]. Therefore, it is important to understand the activation of genes that are responsible for this development [5,39]. Most of the studies, which analyzed gene expression during inner ear development, were performed in animals because the access to fetal human inner ear tissue is still highly limited [39,40,41,42]. The quality and quantity of extracted RNA and the use of suitable reference genes are crucial for gene expression studies. 

Therefore, we compared the RNA extraction methods using FFPE fetal human inner tissue with fresh, frozen tissue stored at −80 °C in RNAlater solution. Lee et al. performed the first extraction of RNA from FFPE inner ear tissue in 1997 [8]. They extracted RNA and detected actin in 1 out of 10 archived FFPE inner ear samples. Lee et al. concluded that RNA extraction from archival sections is limited due to the RNase activity. Two years later in 1999, Ohtani et al. [43] isolated usable RNA from 79% of the FFPE temporal bones and analyzed α-tubulin expression. They concluded that the primer design and their FFPE RNA extraction method was successful. However, using their protocols, we were not able to extract a sufficient amount of RNA from inner ear tissue, which was also partially degraded. FFPE stored tissue is one of the most available resources for molecular biological analysis after histological examination. However, gene expression analyses with FFPE tissue are challenging due to the extraction workflow of decalcification, heating and formalin fixation during the embedding process, which leads to strand breaks and the cross-linking of RNA [3,11,44]. A number of previous studies have attempted to improve the quality of extracted RNA from FFPE stored tissue. Patel et al. (2017) [10], Landolt et al. (2016) [11], and Hamatani et al. (2006) [3] described an approach to improving RNA isolation from archival FFPE stored cancer or renal tissue. Marczyk et al. (2019) [45], compared different RNA isolation methods for fresh formalin-fixed, stored FFPE and fresh RNAlater-stored breast cancer tissue. They concluded that fresh RNAlater-stored tissue yields the highest amount and quality of RNA. In our study, we evaluated two methods using fetal inner ear fresh tissue stored in RNAlater. Like Marczyk [45], we received higher amounts of RNA with RNAlater-stored tissue than with FFPE tissue. As suggested by Hong et al. (2015) [14], we compared the Trizol RNA extraction with a combination of Trizol/RNeasy spin column purification. The combination of Trizol and RNeasy resulted in the highest amount of RNA, with the best RNA quality. Storage in RNAlater solution and the immediate freezing of the tissue led to reduced RNase activity, which resulted in higher amounts of RNA [13,14,46]. RNA isolation from RNAlater-stored tissue is used for several other tissue types, like in bile [47], the cerebellum [48], cartilage [49], serum/plasma [50] or saliva [51], to obtain reliable expression results. We decided to use the combination of Trizol/RNeasy for RNA extraction from RNAlater-stored fetal human inner ear tissue. 

The second aim of our study was to validate stable reference genes for RT-qPCR during fetal development of the human inner ear. Reference genes differ between tissues and developmental stages [52], and no systematic study for selecting appropriate reference genes in the human inner ear at different developmental stages has been performed. Each study suggests the use of a different specific stable gene depending on the tissue. Due to this, we used fetal human material to investigate the most stable reference genes. Therefore, we tested a number of genes from a commercially available human gene panel, which contained the 12 most often used reference genes for RT-qPCR applications, and analyzed their stability with RT-qPCR and RNAseq. Some differences between the RT-qPCR and RNAseq results in the expression levels of the reference genes were observed, although a high correlation between these two methods was reported in different studies [53,54]. The analysis of gene expression from RNAseq data requires a complex computational analysis that includes alignment, quantification, normalization and differential expression analysis [55]. However, the results from different RNASeq analyses and RT-qPCR provide a good correlation for gene expression quantification for genes with medium expression levels, whereas major differences in expression values are detected in genes with high or low expression levels [56]. A different expression profile for RT-qPCR and RNAseq during inner ear development was observed for RRN18S, which exhibited a much higher expression level with a Ct value < 10 than the other reference genes. All other reference genes revealed Ct values between 18.27 (RPLP) and 25.76 (HPRT1), as recommended for RT-qPCR analysis [52].

After performing several algorithms to determine the best reference genes, we detected four reference genes. *B2M*, *GAPDH*, *GUSB* and *HPRT1* were the most stable genes throughout our expression analysis of the developing inner ear. In the expression stability analyses using several algorithms, the NormFinder and the CV analysis revealed that *HPRT1* was the top-ranked gene with the best stability during the investigated time points. By contrast, the pairwise ΔCT showed that *GAPDH* was the top-ranked gene with the lowest average standard deviation and that *B2M* was the gene with the lowest standard deviation in the mean ± SD calculation. These genes are involved in metabolic pathways (*GAPDH*, *GUSB*, *HPRT1*) or in immune response (*B2M*) (OMIM entrez: 138400, 308000, 611499 and 109700). We analyzed the influence of the four selected reference genes on the expression profile of *OTOF* and *TECTA*, two deafness-related genes. We normalized the target genes *OTOF* and *TECTA* with *B2M*, *GAPDH*, *GUSB* and *HPRT1*. The selected reference genes did not differ from each other in their expression profiles; however, their level of expression was different. This difference in expression level may lead to a different interpretation of the expression at several time points. Our analyses showed that different reference genes revealed no stable expression during the development of the human inner ear. Using one single reference gene is generally not recommended, as discussed in Vandesompele et al. (2002) [57], and Panina et al. (2020) [58]. Normalization against one reference gene may lead to misleading results. Therefore, our results demonstrate the use of different reference genes to analyze gene expression in the developing human inner ear.

Our RT-qPCR RNA expression profile for *OTOF* and *TECTA* during inner ear development was confirmed on the protein level by immunohistochemical staining. Both methods revealed an increase in *OTOF* and *TECTA* from developmental stage 12 to 19. Changes in RNA expression have to be confirmed on the protein level since the direct comparison between mRNA and protein levels does not always correlate [59]. Studies in human tissues have shown that the RNA and protein levels of different reference genes are poorly correlated [60], which emphasizes the importance of confirming RNA gene expression data on the protein level. 

## 4. Materials and Methods

### 4.1. Ethical Approval of Fetal Specimens

Specimens (between GW11 and 19) were provided by the UCL London and Newcastle branches of the HDBR: Joint MRC/Wellcome Trust (grant# MR/R006237/1) Human Developmental Biology Resource (http://hdbr.org). Fetal and embryonic tissue was collected, with informed consent, and distributed to research projects under ethical approval 18/NE/0290 from the North-East-Newcastle and North Tyneside 1 Research Ethics Committee for HDBR Newcastle and 18/LO/022 from the Fulham Research Ethics Committee for HDBR UCL London. Specimens were certified by embryologists to exhibit no visible malformations, and their embryological ages were differentiated by quantifying characteristics like the crown–rump length, external and internal morphology and the estimated gynecological age. All specimens were devoid of any external or internal congenital defects. 

### 4.2. Tissue Preparation, Histology and Immunohistochemistry of Paraffin Sections

The tissue preparation for paraffin embedding, the immunohistochemistry procedure and the digital examination of human cochlear sections were described in detail in our previous publications [61,62,63,64,65]. Negative controls were acquired by substituting the primary antibodies with isotype-matching immunoglobulins. These negative controls did not yield any immunostaining. Immunohistochemistry was performed utilizing a Ventana Roche XT immunostainer (Mannheim, Germany), applying a DAP-MAP discovery research standard procedure. Then, 5 µm thick human inner ear FFPE sections were incubated for 1 h at 37 °C with primary antibodies, followed by 1 h at 37 °C with Universal Secondary Antibody (supplied from Ventana, Roche, Mannheim, Germany, 760-4250). The primary antibodies were *OTOF* (polyclonal, rabbit, dilution 1:200, Invitrogen, Karlsruhe, Germany, PA5-79776) and *TECTA* (polyclonal, rabbit, dilution 1:150, Invitrogen, PA5-80102). 

### 4.3. RNA Isolation and Measurement of RNA Quality and Quantity

The tissue preparation for RNA extraction was performed with human fetal inner ear tissue that was paraffin embedded and stored for six months to one year or with fresh, frozen tissue stored at −80 °C in RNAlater solution (Invitrogen, AM7024). Four commercially available RNA extraction kits were used. Protocol No. 1: High Pure FFPET RNA Isolation Kit, Roche Life Science, Mannheim, Germany, Cat. No. 06 650 775 001; Protocol No. 2: RecoverAll^TM^ total Nucleic Acid Isolation Kit for FFPE, Ambion, Ref: AM1975; Protocol No. 3: RNA isolation with Ambion Trizol, Invitrogen, Cat. No. 15596018; Protocol No. 4: published by Hong et al., 2015 [14], with a method involving a combination of Ambion Trizol, Invitrogen, Cat. No. 15596018 and RNeasy Micro Kit, Qiagen, Hilden, Germany, Ref: 74004. Generally, each extraction process included the homogenization of the tissue, protease digestion, binding to solid substrate, washing and elution with specific variations in each protocol. In total, 50 biological replicates (FFPE and fresh tissue) were used for the experiments. The purification of RNA was performed by following the DNA and RNA precipitation manual from Genelink (Catalogue No. 40-5135-05, Elmsford, NY, USA) with Ammonium Acetate. 

The RNA quantity was measured with a BioPhotometer plus (Eppendorf, Hamburg, Germany) and Qubit Fluorometric Quantification (ThermoFisher, Karlsruhe, Germany). The RNA purity was determined with tolerated A260/280 and A260/230 absorbance ratios. Intact 18S and 28S rRNA was measured with the Bioanalyzer 2100 from Agilent (Santa Clara, CA, USA).

### 4.4. cDNA Synthesis and Quantitative Real-Time PCR

First-strand, complementary cDNA was synthesized with the Superscript™ IV VILO Master Mix with exDNAase (Invitrogen, 11766050) using a ThermoQ BioER programmable incubator (BioER Technology Co., Ltd., Hangzou, China) by following the manufacturer’s instructions. For cDNA synthesis, 12 µL of each RNA sample was used because the RNA concentrations varied between 50 ng/µL and 200 ng/µL depending on the sample. The following RT-qPCR was performed with an adjusted cDNA amount of 12.8 ng for each sample using the SensiFAST^TM^ SYBR^®^ and Fluorescein Kit (Bioline, Memphis, TN, USA, BIO-96020) in a final volume of 22 µL (0.9 µL of forward primer, 0.9 µL of reverse primer, 11 µL of 2× SybrGreen, 2 µL of cDNA, 7.2 µL of aqua dest.) in the MyiQ™ Single Color Real Time PCR Detection System (Biorad, Hercules, CA, USA). The RT-qPCR run conditions were 95 °C/2 min, 95 °C/5 s denaturation, 60 °C/10 s annealing and 72 °C/5 s extension for 49 cycles. Melting curve and peak analyses were performed automatically at the end of the RT-qPCR procedure. 

### 4.5. Reference Gene Selection and Primer Design

Twelve potential reference genes were chosen from the commercially available Human Reference Gene Panel (Panel A101, TATAA Biocenter AB, Saarbrücken, Germany; listed in Table 3). To determine the influence of the reference gene, we analyzed the expression of the *OTOF* and *TECTA* genes in the developmental inner ear. The primer sequences for the RT-qPCR of *OTOF* and *TECTA* were obtained from Primer3 (version 4.1.0, https://primer3.ut.ee, accessed on 11 January 2021). Primers were tested for their specificity and dimer formation at appropriate cDNA concentrations. *TECTA* primer forward: 5′-AGTTCTCCTACACCCTCCTG-3′, reverse: 5′-TGCCTCCTATCTTGACCTCC-3′ with amplicon length of 147 base pairs (bp) and primer length of 20 bp. *OTOF* primer forward: 5′-TCCTCAACCCTCTCAAGTCC-3′, reverse: 5′-AGCTTTTTGACCATGTAGCC-3′ with amplicon length of 168 bp and primer length of 20 bp.

### 4.6. RT-qPCR Data Analysis and Statistics

To test the expression stability, we performed NormFinder analysis (version 0.953, https://www.moma.dk/software/normfinder, accessed on 20 June 2022), CV analysis, pairwise ΔCt analysis and a calculation of the mean ± SD in Microsoft Excel 2016. For CV analysis, the Ct values were transformed to linear scale by calculating the 2^−Ct^ for each sample. To assess statistical differences in the RNA quantities between the groups and gestational weeks, one-way ANOVA was performed in Graph Pad Prism 8 (La Jolla, CA, USA). The normalized gene expression of *OTOF* and *TECTA* was calculated as the geometric mean [66]. 

### 4.7. Next Generation Sequencing and Data Analysis

The same RNA samples used for the RT-qPCR with a high RIN (ratio of 28S to 18S rRNA) quality (RIN > 7) were chosen for the NGS. 3′ mRNA sequencing libraries were created using the Quant Seq 3′ mRNA-Seq Library Prep Kit (Lexogen, Wien, Austria). Finally, RNA sequencing was performed using an ION Proton platform (Thermo Fisher, Karlsruhe, Germany) according to the manufacturer’s instructions, yielding 7–8 million reads per sample. The raw RNAseq data were pre-processed using the nf-core/rnaseq/Nextflow pipeline (version 3.9) [67,68]. The reads were aligned to the hg38 reference genome using STAR [69], and the gene expression was quantified with RSEM [70]. Differentially expressed genes (log2 fold change > |1|, FDR < 0.5) were identified using the Bioconductor R package DESeq2 (version 4.2 [71]). Custom R scripts for generating plots were used for the analysis and visualization (all codes are available from the corresponding author upon reasonable request) of RT-qPCR and the RNAseq log2 fold change expression.

## Figures and Tables

**Figure 1 ijms-25-02907-f001:**
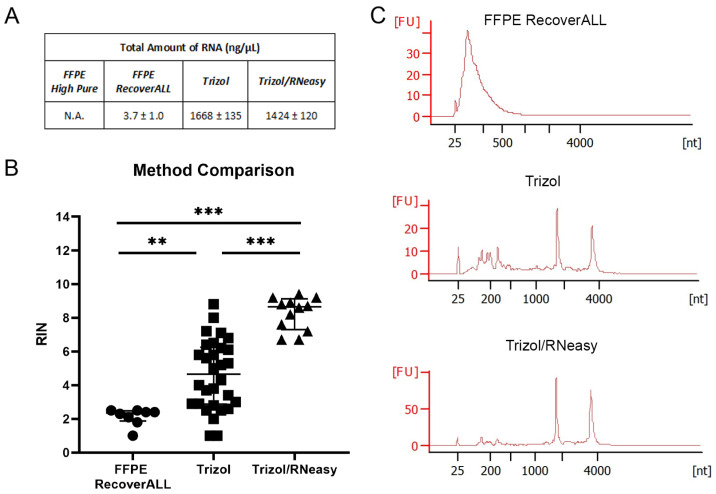
Comparison of four different RNA extraction methods for human fetal inner ear tissue. (**A**) Total mean ± SD amount of RNA obtained with the different extraction methods. (**B**) RNA integrity number (RIN) and values of three RNA extraction methods plotted as dot plots with interquartile range (10 = intact; 0 = degraded RNA); one-way ANOVA with Tukey’s multiple comparisons test; ** *p* < 0.01; *** *p* < 0.001. (**C**) Electropherogram of FFPE RecoverALL method, fresh frozen tissue Trizol method and fresh frozen tissue Trizol/RNeasy method. The 18S and 28S rRNA peaks were clearly visible with the combination method Trizol/RNeasy. FFPE RecoverALL n = 8, Trizol n = 31, Trizol/RNeasy n = 11.

**Figure 2 ijms-25-02907-f002:**
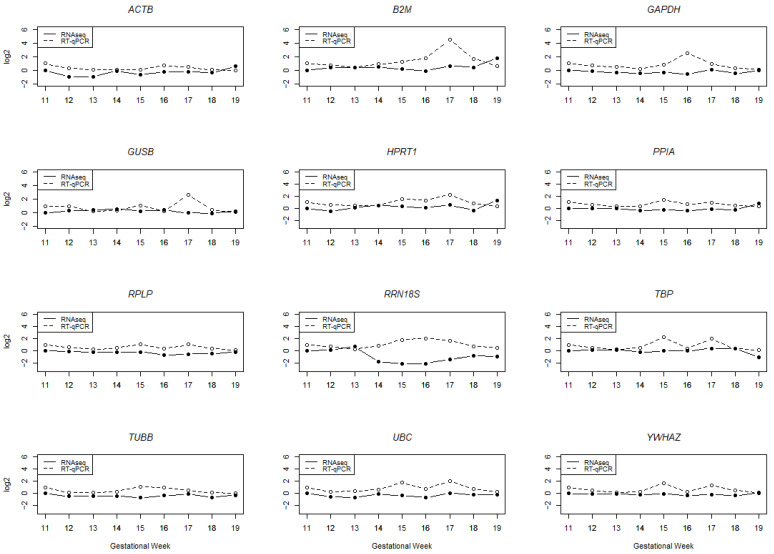
RNAseq and RT-qPCR raw expression profile of 12 reference genes. GW11 is taken as the experimental calibrator for RT-qPCR and RNASeq data. The RNAseq profiles are represented as DESeq2 log2 fold changes in expression for all gestational weeks. The RT-qPCR expression profile is shown as the log2 fold change with the geometric mean.

**Figure 3 ijms-25-02907-f003:**
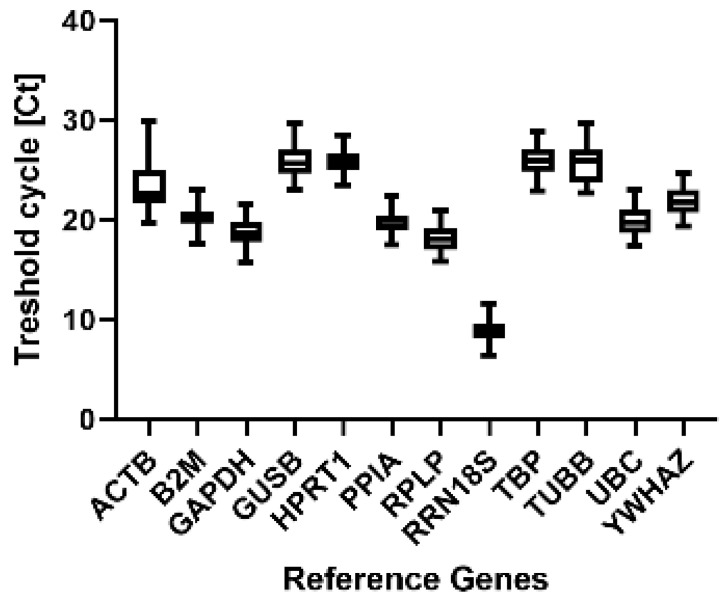
RT-qPCR cycle threshold (Ct) values of 12 candidate reference genes at different developmental time points in the human inner ear. Graphical representation of Ct values as box plots for all reference genes. The boxes extend from the 25% to the 75% percentile. The black bars denote the median and the whiskers delimit the 1.5-fold interquartile range.

**Figure 4 ijms-25-02907-f004:**
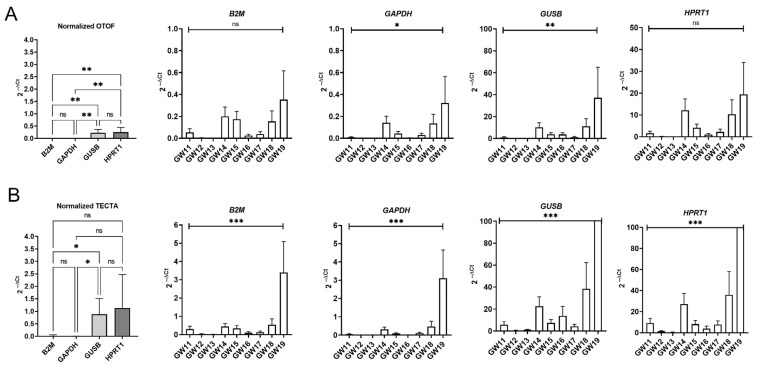
RT-qPCR of the target genes *OTOF* (**A**) and *TECTA* (**B**) with different reference genes. Both target genes were normalized with *B2M*, *GAPDH*, *GUSB* and *HPRT1.* Overall expression of both genes with reference genes (left graphs) and expression profiles during development between gestational weeks 11 to 19 (right graphs). One-way ANOVA with the Brown–Forsythe test was performed: * < 0.05; ** < 0.01; *** < 0.001; ns = not significant.

**Figure 5 ijms-25-02907-f005:**
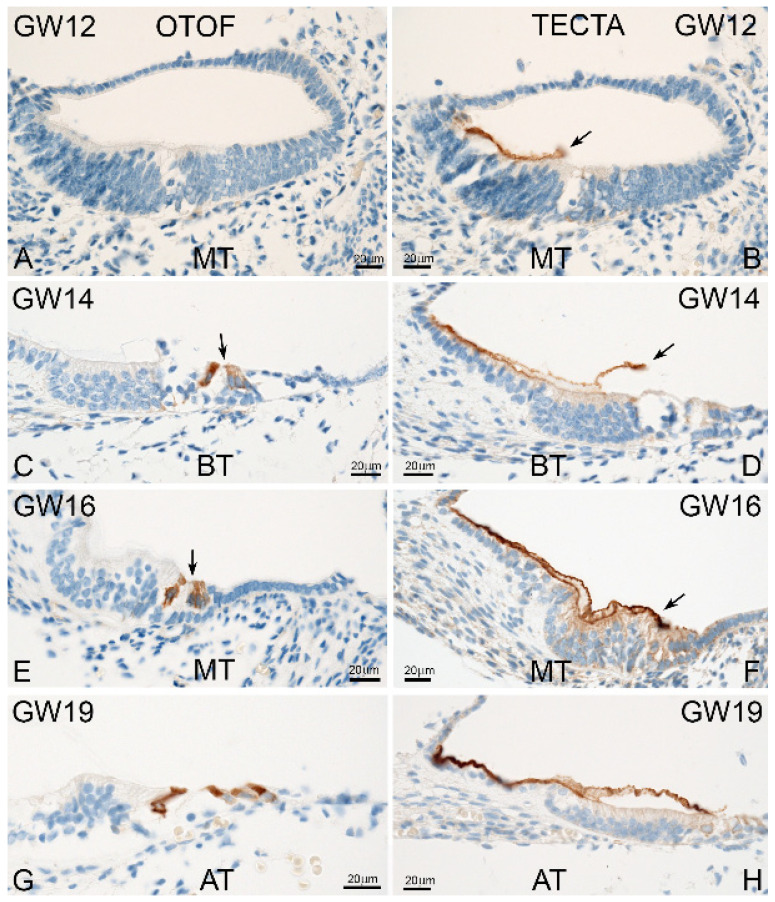
Immunostaining of *OTOF* and *TECTA* from the human fetal developmental stages GW12 to GW19. (**A**,**C**,**E**,**G**) First *OTOF* staining was observed at GW14. The outer hair cells (OHCs) revealed weaker staining (arrow). At GW16 up to GW19, OHCs and inner hair cells (IHCs) displayed the same staining intensity for *OTOF*. (**B**,**D**,**F**,**H**) The immunostaining of *TECTA* was visible in the tectorial membrane (arrow) at all investigated stages (GW12 to GW19). AT: apical turn, MT: middle turn, BT: basal turn.

**Table 1 ijms-25-02907-t001:** Brown–Forsythe one-way ANOVA was performed to analyze differences between the group means of the RT-qPCR data. * *p* < 0.05; ** *p* < 0.01; *** *p* < 0.001; ns = not significant.

	Statistically Significant			
	***	**	*	ns
Genes	*B2M* *GUSB*	*GAPDH* *HPRT1* *TBP*	*ACTB* *TUBB* *UBC* *YWHAZ*	*PPIA* *RPLP* *RRN18S*

**Table 2 ijms-25-02907-t002:** Expression stability of different reference gene analyses with NormFinder, mean ± SD, CV analysis and pairwise ΔCt.

	NormFinder			Mean ± SD			CV Analysis		Pairwise ΔCt	
Rank	Gene	Stability S	SD	Gene	MV	SD	Gene	Stability M	Gene	Average SD
1	*HPRT1*	0.027	0.005	*B2M*	20.36	1.1	*HPRT1*	80.88	*GAPDH*	1.994
2	*TBP*	0.029	0.005	*PPIA*	19.84	1.13	*PPIA*	82.48	*TUBB*	2.069
3	*PPIA*	0.035	0.006	*HPRT1*	25.76	1.17	*RPLP*	85.72	*RRN18S*	2.168
4	*RPLP*	0.037	0.006	*RRN18S*	8.96	1.21	*UBC*	88.68	*TBP*	2.306
5	*UBC*	0.040	0.007	*UBC*	19.97	1.31	*RRN18S*	90.34	*GUSB*	2.307
6	*YWHAZ*	0.043	0.007	*RPLP*	18.27	1.38	*YWHAZ*	95.8	*RPLP*	2.330
7	*B2M*	0.043	0.007	*TBP*	25.91	1.38	*B2M*	96.84	*ACTB*	2.335
8	*TUBB*	0.044	0.007	*GAPDH*	18.78	1.41	*GUSB*	99.09	*HPRT1*	2.378
9	*GUSB*	0.050	0.008	*YWHAZ*	21.98	1.61	*TBP*	106.96	*PPIA*	2.407
10	*GAPDH*	0.053	0.008	*GUSB*	25.74	1.68	*GAPDH*	108.11	*B2M*	2.469
11	*ACTB*	0.088	0.013	*TUBB*	25.58	1.96	*TUBB*	108.28	*YWHAZ*	2.595
12	*RRN18S*	0.105	0.015	*ACTB*	23.22	2.55	*ACTB*	128.32	*UBC*	2.663

**Table 3 ijms-25-02907-t003:** List of reference genes used in this study with their cellular function.

Full Name	Gene	Function	PCR Amplicon Length	Accession No.
β-Actin	*ACTB*	Component of cytoplasmic cytoskeleton	188 bp	HGNC:132
18S ribosomal RNA	*RRN18S*	Component of small subunit 40S of eukaryotic ribosome complex	120 bp	HGNC:44278
Tubulin, beta polypeptide	*TUBB*	Structural component of microtubules	119 bp	HGNC:20778
Glyceraldehyde-3-phosphate dehydrogenase	*GAPDH*	Has role in glycolysis for catalyzing D-glyceraldehyde 3-phosphate into 1,3-bisphospho-D-glycerate	151 bp	HGNC:4141
β-2-Microglobulin	*B2M*	Component of MHC1 complex, involved in peptide presentation to the immune system	161 bp	HGNC:914
60S acidic ribosomal protein P0	*RPLP*	Large subunit of eukaryotic ribosomes	150 bp	HGNC:10371
TATAA-box binding protein	*TBP*	Initiation of transcription by RNA polymerase II	174 bp	HGNC:11588
β-Glucuronidase	*GUSB*	Hydrolase for degradation of glycosaminoglycan, localized to lysosomes	165 bp	HGNC:4696
Hypoxanthine-guanine phoshoribosyltransferase 1	*HPRT1*	Catalyzes conversion of hypoxanthine to inosine monophosphate and guanine to guanosine monophosphate via transfer of the 5-phosphoribosyl group from phosphoribosyl 1-phyrophasphate; Has role in generation of purine nucleotides in purine salvage pathway	94 bp	HGNC:5157
Peptidyl-propyl isomerase A, Cyclophilin A	*PPIA*	Catalyzes cis-trans isomerization of proline imidic peptide bonds in oligopeptides and the accelerated folding of proteins	114 bp	HGNC:9253
Ubiquitin C	*UBC*	Ubiquitination in several cellular functions	240 bp	HGNC:12468
Tyrosine 3/tryptophan 5-monooxygenase activation protein, zeta polypeptide	*YWHAZ*	Mediates signal transduction by binding to phosphoserine-containing proteins	248 bp	HGNC:12855

## Data Availability

Data are contained within the article.

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
