# Peer review of "Validation of RNA Extraction Methods and Suitable Reference Genes for Gene Expression Studies in Developing Fetal Human Inner Ear Tissue"

_ijms, 2024, doi:10.3390/ijms25052907_

Round 1

Reviewer 1 Report

Comments and Suggestions for Authors

All sugestions and comments are icluded in the attachment. 

Comments on the Quality of English Language

Minor editing of English language required

Reviewer 2 Report

Comments and Suggestions for Authors

 General comments

If the techniques have been used in other parts of the fetal body, such as the liver or the heart, why should the inner ear be examined? Why is this necessary if the technique has worked before? 

Please be more specific, why have you chosen these genes? Is there animal data, what is the function of the gene? From when is it expressed in animals? This is much more interesting,

Please show data on the expression of these genes in other tissues? like brain or liver? Perhaps their threshold changes are similar? So the question is, are these genes really relevant?

Please also add in the introduction where the fetuses are from, is this a database? When were they collected? 

How can you use mouse as a positive control? There are many ABs that bind in mouse and not in human? Please show the pictures of mouse staining at the same gestational weeks, this would be interesting for comparison. 

Specific comments.

1.     Introduction

L 42:                 major interest

L 42:                 wording: It is of major interest to discover the genes that are regulated during the development and the consequences of these genes being disrupted or absent, to understand the background of inner ear hearing loss.

LL 51 – 54         structure: Please modify the structure of the sentence for a better understanding.

Please add a paragraph concerning the RNA-Protein-Correlation of the inner ear to highlight the high relevance of the RNA Level in inner ear development. E. g. preclinical studies.

Please add a paragraph concerning the potential therapeutical relevance of high quality RNA. E. g. future clinical application genome editing.

Please add a paragraph concerning OTOF and TECTA (see Ll 150). This i spart oft the introduction, not of the results.

2.     Results

L 79:                 Fig. 1.: n = 50

L 81:                 500 ng

L 90:                 TRIzol

Ll 102 - 106:     Part of the introduction

Fig. 3:               Please add explanations of the boxplots in text.

Fig. 5:               Please add explanation for MT, BT, MT and AT in text/legend of figure 5.

3.     Discussion

L 176:               Please add reference.

L 178 – 180:      Please add reference.

L 215:               most stable reference genes

L 217:               reference genes

4.     Materials and Methods

Ll 251 – 255:     Please explain the relevance for this part of the text.

L 258:               Digital examination

L 281:               Please add tolerated A260/280 and A60/230 Ratios.

Please add necessary information following MIQE guidelines (Kriterion E):

Bustin, S. A., Benes, V., Garson, J. A., Hellemans, J., Huggett, J., Kubista, M., Mueller, R., Nolan, T., Pfaffl, M. W., Shipley, G. L., Vandesompele, J., & Wittwer, C. T. (2009). The MIQE guidelines: minimum information for publication of quantitative real-time PCR experiments. Clinical chemistry55(4), 611–622. https://doi.org/10.1373/clinchem.2008.112797

Reviewer 3 Report

Comments and Suggestions for Authors

The novel component of this article is the use of specialised tissue – human foetal tissue. It noted that this is very difficult to access this type of tissue for medical research and thus, is extremely valuable. Thus, I gave a high ranking for "novelty" rating. 

The authors have highlighted issues with RNA extraction that are well known in the molecular field.

The authors demonstrate that the deciding what reference gene(s) used is incredibly important and thus, impacts the outcome of gene expression data. Unfortunately, the manuscript needs major revision as there are too many spelling mistakes, methodlogy is incomplete to reproduce, and the images are hard to decipher or are confusing to interpret. On a better note, the authors have presented high quality images for the immunohistochemistry data of the ear which has its own issues with sectioning (assuming it's mouse).

Specific corrections:

RT-PCR stands for reverse transcription. Use the acronym “qRT-PCR” instead. 

Line 18

“RNA frequently used to determine the spatio-temporal expression of genes, which are regulate during the development of the human inner ear.”

Sentence does not make sense. Recommend deleting this sentence.

Gene names must be in italics; OTOF, TECTA – e.g.  missing from Lines 29 and 30 

Line 23

It is unclear how old the paraffin embbed tissue was (ie. Decades, years, or months). Having an indication of age of the block would be helpful even if it is a time range -add this to the methods. 

Line 79

Separate out the number of samples per group or add directly to bar graph.

Lines 99, 178, 295 – Use of English spelling’- analyses vs. American spelling, analyses. Keep format consistent.

Figure 1A

Add “RNA” to the first row in the table. Ie Total Amount of RNA (ng/ul)

Figure 2 – improve the resolution of the text in the image.  Add the key to the figure legend.

Line 111 – write acronym in full for GW.

Figure 2 

Arrangement of genes – use alphabetical order or according to grouping – no change vs change.

Figure 3 

– label x-axis with “reference genes”

– correct spelling mistake – Threshold. 

_ order of genes listed – make alphabetical. 

Figure 4:

Improve resolution of the y-axis label

Improve the positioning of the *** horizontal lines for HPRT1, GUSB

List genes in alphabetical order; B2M, GAPDH. GUSB, HPRT1

Line 155

Remove (APA style)

Line 162

Unclear what tissue was examined – assuming it the mouse ear? Animal ethics approval?

Figure 5

What do the “MT” and “AT” stand for?

Line 234

Spelling mistake – “discussed”

Lines 240, 337, 339

Spelling mistake - feotal 

Line 251

Delete “Research manuscripts reporting large datasets that are deposited 251

in a publicly available database should specify where the data have been deposited and 

provide the relevant accession numbers. If the accession numbers have not yet been obtained at the time of submission, please state that they will be provided during review. 

They must be provided prior to publication.”

Line 256

Include in this section the size of sections examined – 5 uM?

Were the antibodies checked by western blot analysis for the detection of the correct molecular weight expressed protein. 

What was the concentration of the secondary antibody and detection method used. 

Line 259

Age of mouse specimens for positive control samples. 

Line 288

What was the starting RNA concentration you used to prepare the cDNA?

Table 3

Do the PCR primers target exon-intron boundaries or exon only?

Were the RTPCR amplicons sequence to confirm gene identity?

Line 297

Harvard Primer Bank – checking to see if existing primers have been designed and published. Please acknowledge if you used primers from this website. 

Line 305

RTPCR conditions are not listed. What Taq, Primer concentration, cycle number. 

Comments on the Quality of English Language

Too many spelling mistakes. Some text is poorly written or they have not taken out the manuscript instructions. I think they may have submitted the incorrect version of the manuscript.

Round 2

Reviewer 3 Report

Comments and Suggestions for Authors

I am extremely satisfied with the answers the authors have prepared. They have addressed all my concerns. In addition, with incorporation of my changes and the second reviewer, the manuscript has enhanced its scientific rigour and readability/reproducibility for the end-user.